# Functional gene categories differentiate maize leaf drought-related microbial epiphytic communities

Barbara A. Methe[1,2], David Hiltbrand[3], Jeffrey Roach[4], Wenwei Xu[5], Stuart G. Gordon[6], Brad W. Goodner[7], Ann E. Stapleton[3]*

1 J Craig Venter Institute, Medical Center Drive, Rockville, MD, United States of America, 2 Department of Medicine, University of Pittsburgh, Pittsburgh, PA, United States of America, 3 Department of Biology and Marine Biology, University of North Carolina Wilmington, Wilmington, NC, United States of America, 4 Research Computing, University of North Carolina Chapel Hill, Chapel Hill, NC, United States of America, 5 Agricultural and Extension Center, Texas A and M AgriLife Research, Lubbock, TX, United States of America, 6 Biology Department, Presbyterian College, Clinton, SC, United States of America, 7 Department, Hiram College, Hiram, OH, United States of America

* stapletona@uncw.edu

**Data Availability Statement:** All relevant data are uploaded to the Dryad repository and publicly accessible via the following DOI: 10.5061/dryad.

## Abstract

The phyllosphere epiphytic microbiome is composed of microorganisms that colonize the external aerial portions of plants. Relationships of plant responses to specific microorganisms–both pathogenic and beneficial–have been examined, but the phyllosphere microbiome functional and metabolic profile responses are not well described. Changing crop growth conditions, such as increased drought, can have profound impacts on crop productivity. Also, epiphytic microbial communities provide a new target for crop yield optimization. We compared *Zea mays* leaf microbiomes collected under drought and well-watered conditions by examining functional gene annotation patterns across three physically disparate locations each with and without drought treatment, through the application of short read metagenomic sequencing. Drought samples exhibited different functional sequence compositions at each of the three field sites. Maize phyllosphere functional profiles revealed a wide variety of metabolic and regulatory processes that differed in drought and normal water conditions and provide key baseline information for future selective breeding.

## Introduction

Plants form a wide variety of intimate associations with a diversity of microorganisms in the phyllosphere, the above-ground plant surface [1, 2]. Microorganisms can exist as endophytes within the plant, as epiphytes on plant surfaces (which together compose the phyllosphere) and in the soil surrounding and in the roots [3–5]. The ubiquity and intricacy of these plant-microbe associations support the model of the plant as a"meta-organism" or"holobiont" consisting of the host and its microbiome (the collection of microorganisms and their gene content) which maintain a relationship over the lifetime of the plant [3, 6, 7]. The plant-associated microbiome, the phytobiome, is a complex and dynamic system existing as both an agonist

7m0cfxprs. https://datadryad.org/stash/dataset/doi:10.5061/dryad.7m0cfxprs

**Funding:** This material is based upon work supported by the National Science Foundation under Grant No. 1126938 to AES and BM. The funders had no role in study design, data collection and analysis, decision to publish, or preparation of the manuscript.

**Competing interests:** The authors have declared that no competing interests exist.

and antagonist of plant fitness and adaptability [8, 9]. Therefore, elucidating the nature and extent of these interactions offers significant opportunities for improving plant health, for example, through alterations in nutrient cycling, neutralizing toxic compounds, discouraging pathogens, and promoting resistance to abiotic stresses that have the potential for generating significant impact on plant productivity [10–14]. Optimization of selective breeding for epiphytes presents new challenges in ensuring that microbe colonization occurs as needed, while presenting new potential effective indirect genetic selection [14–16] for crop improvement. Ultimately, engineering microbial and plant genotypes for optimal function and resilience will also require causal, mechanistic analyses of gene and pathway level processes; one first step in such mechanistic analysis for the microbial components of the phyllosphere is construction of controlled synthetic communities of microbes or assembly of specific sets of microbial functional genes [17].

In contrast to the rhizosphere, the region of soil that is directly influenced by root secretions, the phyllosphere is both a relatively understudied and transitory microbial environment [2, 18]. Microbial epiphytes of the phyllosphere experience an environment subject to different influences than those found in the rhizosphere and from host endophytes. Those in the phyllosphere experience atmospheric influences including direct sunlight exposure during diurnal cycles, and barriers such as waxy cuticle resulting in an oligotrophic environment [19, 20]. More labile associations between epiphytic microbes and host leaves do present an opportunity for interventions. For example, inoculation of beneficials or application of probiotics [21] could be done rapidly, during crop growth, since above-ground leaves and stems are easy to access. Longer term interventions such as selection of host genotypes that support specific desired microbial functions on external leaf surfaces at key points during growth or in response to biotic or abiotic stress could also be attempted [22, 23].

Corn, *Zea mays* L., is a widely grown and economically important annual crop. Drought is an abiotic stress that can negatively affect plant productivity [24]. Hence, understanding the role that the phyllosphere may play in association with maize undergoing abiotic stress is a priority. Epiphytic microbes are a unique target for drought tolerance. Targeting such microbes has potential advantages in the speed of alterations relative to plant breeding. It also provides the potential for temporal targeting through inoculation only during the adverse conditions [14]. Supporting this potential, seed microbial inoculation for crop drought tolerance is already in commercial use (for example, https://www.indigoag.com/).

Only a small fraction of microbial diversity is culturable in vitro [25, 26]. This has led to the use of culture independent methods for study of microbial community structure and function. For approximately the past two decades, microbial and fungal diversity has been described via the sequencing of amplicons representing biomarkers, such as the 16S rRNA gene (bacteria) and internally transcribed spacer (ITS) regions (fungi) [27, 28].More recently, techniques in microbial community research have shifted to investigate community structure and function at a systems-wide level. One such systems method, metagenomics, involves sequencing and analyzing genes derived from whole communities as opposed to individual genomes. Examining microbiomes at this level has shown that microbes ultimately function within communities rather than as individual species [9]. The traditional use of taxa to investigate microbiomes does not fully account for metabolic interactions between species. Typically functional genes exhibit different patterns than taxa, and functional genes are often better predictors of niche [29–31]. In addition, functional gene content can be more heritable (i.e., more driven by host genetic interactions) [32]. Functional gene analyses also provide key information needed for community-level metabolic engineering [14, 22].

To address our questions about functional differences between microbial communities, we selected a factorial design with use of multiple field sites to increase generality. We know that

plant breeding requires consideration of environmental contributions. By prioritizing multiple field sites in our initial investigations, our results provide critical information for future experimental designs for breeding and extension of the experiments found here.

### Seed stocks

*Zea mays* L. inbred B73 seed was supplied by the Maize Stock Center, http://maizecoop.cropsci.uiuc.edu/, and seed was increased at the Central Crops Agricultural Research Station, Clayton, NC using standard maize nursery procedures. Genotype of the seed lots used for these experiments was verified by SSR genotyping using eleven markers and comparison of fragment sizes to the sizes listed in the MaizeGDB database [33], http://www.maizegdb.org/ssr.php.

### Experimental design and field sampling

Research field sites were generously provided by collaborators with ongoing scientific and extension projects; no additional permits or permissions were required. For this experiment we used a hierarchical design, with the treatment plots nested in each field site. There were three randomly arranged plots within each treatment level at each field site, surrounded by additional plant plots. Replicated field plots were planted in Albany, CA at the USDA University of California-Berkeley field site (abbreviated as CA), 37 degrees 53 min 12.8 sec N 122 degrees 17 min 59.8 sec W, on June 6, 2012. The field site had uniform soil and subsurface irrigation and fertilization supplied according to normal agronomic practice for this growing area. The southern section had normal irrigation throughout the season. However, the northern section had normal irrigation until vegetative growth stage V5 when all watering was stopped for two weeks; after sampling of leaves irrigation was resumed to allow plant growth to maturity. Seeds were planted at two sites in Texas, Dumas Etter field (abbreviated as DE) 35.998744 degrees N 101.988583 degrees W on May 8, 2013, and Halfway, TX field (abbreviated as HF), 34.184136 degrees N 101.943636 degrees W on April 26, 2012. The sites had center-pivot irrigation and standard maize field management. Drought treatment blocks were watered at 75% of the normal rate at DE and at 50% of the normal rate at the HF field site. The DE field site had one replicate plot that experienced additional rain late in the season (after phyllosphere sample collection). The HF field site had no unmanaged precipitation between July 9 and harvest. Late-season (post-phyllosphere sampling) field trait measurement methods and data files for each field site are provided in Supplemental Plant Traits files 1–7 in doi: 10.5061/dryad.7m0cfxprs.

### Field trait measurement

At the Texas field sites (DE and HF) plant and ear heights were measured once per plot when growth was complete after tasseling. Ears were harvested and shipped to UNCW for measurement. For each ear, cob diameter at the base was measured with digital calipers, and twenty seeds were removed from the middle of each cob, placed in envelopes, and weighed. For the CA site, individual plant heights were measured and cobs were collected at the end of the season, October 1–3, 2012. Seed development was not complete, so only cob traits were measured. Cob diameter at base was measured with digital calipers; cob length was measured with a ruler. Plant data for each location and trait are included in doi: 10.5061/dryad.7m0cfxprs as plant trait files 1 to 6, with metadata about the column headers in file 7.

### Leaf sampling and DNA extraction

Samples were collected from DE on June 26, 2012 and from HF on June 27, 2012, at developmental stage V8. The CA phyllosphere samples were taken August 7 and 8, 2012, at developmental

stage V8. Six fully expanded leaves from the top quarter of the plants in each plot were placed into sterile bags (Whirl-Pak, Nasco, Fort Atkinson, WI) prefilled with 300 mL sterile water and 3 microliters Silwet L-77 (EMCO, North Chicago, IL). Bags were moved to nearby shelters, sonicated for one minute to loosen epiphytic microbes, and the 300 mL of wash solution was filtered through sterile Pall microfunnel 0.2 micron filter cups (VWR, Radner, PA) to collect microbial cells on the filter surface. The filter was removed from the cup with sterile tweezers and dropped into small sterile Whirl-Pak bags then stored frozen until DNA extraction. DNA was extracted from each filter with a PowerSoil Mega kit (MoBio, Carlsbad, CA). Samples were concentrated with filter-sterilized sodium chloride and absolute ethanol according to the manufacturer's instructions and shipped frozen to JCVI for sequencing. Supplemental methods video links are available in the supplemental files repository at Data Dryad doi: 10.5061/dryad.7m0cfxprs, to provide additional details on the protocol used for leaf washes and filtering. Mock and soil samples were sampled using the leaf-wash protocol, to allow detection of any sequences likely to have entered the samples from soil particles or air and from our sampling equipment.

## Library construction and sequencing

All library construction and sequencing were completed using Illumina reagents and protocols. Samples PHYLLO09 and PLYLLO10 were sequenced with Illumina HiSeq and all other samples were sequencing using the MiSeq platform. Two nucleic acid negative control filters were also processed through DNA extraction and library construction and sequencing to test for the presence of any significant contamination of experimental samples by exogenous DNA. One sample from HF drought was lost during processing. The raw data and processed reads are accessible from the NCBI Short Read Archive under Bioproject PRJNA297239.

Because of the nature of the sampling collection and nucleic acid procedures, plant host genomic DNA was inevitably included in the nucleic acid samples used for library construction. Therefore, a screening process was implemented to remove both sequencing artifacts and reads most likely to be of maize origin. Adaptor sequences were removed from the SRA sequencing reads using Trim Galore version 0.4.3 <https://www.bioinformatics.babraham.ac.uk/projects/trim_galore/>. The reads were subsequently filtered to remove maize sequences by alignment using bowtie2 version 2.2.9 [34] to v4 of the B73 Zea mays reference, Zm-B73-REFERENCE-GRAMENE-4.0 <ftp://ftp.ensemblgenomes.org/pub/plants/release-37/fasta/zea_mays/dna/Zea_mays.AGPv4.dna.toplevel.fa.gz> [35]. Quality control at each processing step: initial reads, after adapter trimming, and after host filtering, was verified by FastQC v0.11.7. Read pairs that could be joined were joined with vsearch, v1.10.2 linux x86 64, <https://github.com/torognes/vsearch> [36] and all resulting single-end reads: those that joined and those that did not, were retained for further analysis. UniProt50 protein annotation was performed using HUMAnN2 v0.9.1, <https://github.com/leylabmpi/humann2>, [37] resulting in estimates of gene family count, path abundance, and path coverage together with estimates of taxonomic profile at the species level generated by MetaPhlAn2 [38]. Gene family HUMAnN2 output was explicitly normalized to counts per megabase to adjust for different input library sizes. HUMAnN2 is a reference-based method, so we focus on comparisons between drought and control conditions within the experiments (as all reference-based methods rely on existing data). Full details of parameters, software packages, and scripts used to manage analyses are available in Data Dryad repository doi: 10.5061/dryad.7m0cfxprs.

## Count data analysis

Analysis of the number of reads for each UniProt annotation in each sample was performed with ENNB [39]. The parameters and full R scripts for analyzing the data (along with an R

notebook explaining the process) are available in Data Dryad at doi: 10.5061/dryad.7m0cfxprs. ENNB is a two-stage process with an elastic net for feature selection then negative binomial fit to identify significant annotations, though it is only possible to fit one factor (nested or full factorials for multiple experimental factors are not possible to fit using this two-stage multivariate method). The package was downloaded from the An web page (http://cals.arizona.edu/anling/ software.htm) and scripts written to run both method 1, the trimmed mean (TMM) from the EdgeR package, and method 2, DE-Seq-type count overdispersion. Statistical analysis of annotations different in drought and well-watered conditions were carried out for each field site. The simulations that were created as described in the Simulation Construction section below were used to set the P-value threshold for the analysis of the samples. Imputation of samples was used to calculate the lambda value for cross-validation in ENNB, as specified in the ENNB documentation. The multiple imputation function within ENNB was used to create a third HF drought data column, as ENNB required three samples. After analysis, the annotation data sets were cleaned to remove any rows with annotation IDs that were present in the soil or mock-collected sequenced samples. All input files, R code, an R notebook explaining the analysis, and output files are available at doi: 10.5061/dryad.7m0cfxprs.

## Visualization of significant annotations

Uniprot lists were converted to Gene Ontology lists (not a 1 to 1 mapping) using the conversion web tool at EBI, with lists available in the supplemental data in doi: 10.5061/dryad. 7m0cfxprs, then the lists of GO Process and GO Function annotations that were significantly different upon output from ENNB were visualized using REVIGO [40], http://revigo.irb.hr/, with the Simrel and medium list defaults selected. The REVIGO cytoscape-format xgmml network files were color-coded and the network layout redrawn using Cytoscape v3.2.1 [41]. Venn diagrams for comparison of lists were created with http://www.webgestalt.org/GOView/ [42].

## Simulation construction for analysis validation

In order to measure the precision and accuracy of our analysis pipeline, we constructed simulated files of sequences and processed these through our analysis pipeline to generate simulated counts. Then, we analyzed the simulated counts with ENNB and functions to tabulate true and false positives. We modified and updated FunctionSim (https://cals.arizona.edu/ anling/software/FunctionSIM.htm) to generate sequences with signal and noise that were made independently of our real data. The full set of scripts and parameters is available in Data Dryad at doi: 10.5061/dryad.7m0cfxprs. We tested multiple ENNB thresholds for declaring significant annotations to select suitable cutoff and analysis options with the lowest possible false positive rate. The goal for the simulations to determine if ENNB was a viable method for detecting gene counts between groups. We used a threshold alpha of 0.001. The lowest FDR (0.088) calculated using simulation-group comparisons was used to determine that ENNB would be an acceptable tool to run against the real data, provided the sequence match value to declare similarity was set to a suitably high level. The confusion matrices (true and false positives and negatives) for a range of parameter and sequence similarities are available in the Dryad repository; the notebook is genefamilies_simulations.Rmd.

## Statistical analysis of plant traits

Plant traits (seed weight, plant height, and cob diameter) were analyzed with linear regression models using JMP11 Pro (SAS, Cary, NC) with an adjusted alpha of 0.05. Models were fit with water treatment (as a nominal factor) for each trait. For HF and DE cob diameter traits, plot

numbers were used to identify the group of plants within the larger field and those plot IDs were included in the model to account for the blocks. The number of replicates for each comparison is provided in the box plot figure legend.

## Availability of data and materials

Metagenomic sequences are available in the SRA repository, identifier BIOPROJECT PRJNA297239. All data analysis scripts, simulations, intermediate files and metadata files are available from Data Dryad doi: 10.5061/dryad.7m0cfxprs. A preliminary version of this work is available in bioRxiv under bioRxiv 104331 doi https://doi.org/10.1101/104331.

## Results

To examine the microbial metabolic and regulatory functions important for leaf epiphytic community differences between drought and well-watered field plots, we developed a nested experimental design and a per-field-site analysis using factorial multivariate approaches suitable for our zero-inflated annotation read count data. We prioritized comparisons within multiple geographically diverse field sites. Genotype–environment interaction is a key logistic and experimental constraint for future host plant breeding for improved varieties that would support optimal microbial communities.

We saw little relationship between depth of microbial sequence and annotation quality (Table 1).; for example, in comparison of samples 11 and 12 where sequence depth was not correlated to signal level. Both of the soil samples and one mock sample had no sequence signal (Table 1). The second mock sample contained some sequences that were not classified as contaminants. All annotation rows present in the mock sample were removed from all sample rows before statistical analysis.

## Annotations differing between drought and watered treatments

To robustly determine the ENNB parameters with the fewest false positives we created simulations using an independent sequence database. Then, we processed the simulations through our sequence read and statistical analysis code and measured the number of true and false detections. For count analysis, use of the trimmed mean adjustment (Tmm1) and a threshold of 0.001 for negative binomial fitting gave fewer false positives and we report results using those thresholds. Our analyses may be re-run using the scripts and setting information provided in the Dryad repository supplemental files if different P-value thresholds are desired.

Drought and watered plots at each site had significant differences in read counts for regulatory and metabolic functions. The ENNB analysis with normalization by TMM generated a list of significant GO Process and GO Function annotations in watered as compared to drought-treated phyllosphere samples for each field site, with groups of related GO terms from REVIGO analysis indicated by edges between GO node terms. Larger nodes indicate the frequency of the annotation in the GO database, so smaller nodes with no edges such as bacteriocin immunity (Fig 1A) are the most unique. The significant GO Process terms identified as semantically distinct in the drought treatment for the Albany, CA field (abbreviated as CA) site (Fig 1A) include biochemical pathways involved in basic cellular responses, such as transcription and DNA replication, and specific metabolic remodeling pathways, such as isoleucine biosynthesis. Pathways we observed that are likely to be important for microbial community interactions include bacteriocin immunity and amino acid transport [43]. Functional annotations (Fig 1B) for the CA field site are similar to process annotations, with the addition of a cluster of energy-metabolism related binding functions, such as NADP binding (Fig 1B).

**Table 1. Sample characteristics.**

| Sample ID | Field Site[1] | Treatment Type | Sequence Amount[2] | Sequence Comment |
|---|---|---|---|---|
| PHYLLO9 | HF | watered | deep | |
| PHYLLO10 | DE | watered | deep | |
| PHYLLO11 | CA | watered | small | all contaminant |
| PHYLLO12 | CA | drought | large | low proportion of signal |
| PHYLLO13 | CA | watered | large | |
| PHYLLO14 | CA | drought | moderate | |
| PHYLLO15 | DE | watered | moderate | |
| PHYLLO16 | DE | drought | small | |
| PHYLLO17 | CA soil | watered | small | soil sample below watered plot plants, all contaminant |
| PHYLLO18 | CA soil | drought | small | soil sample below drought plot plants, all contaminant |
| PHYLLO19 | mockDE | none | small | |
| PHYLLO20 | mockCA | none | small | low proportion of signal |
| PHYLLO21 | CA | watered | large | |
| PHYLLO22 | CA | drought | large | |
| PHYLLO23 | DE | watered | large | |
| PHYLLO24 | DE | drought | small | |
| PHYLLO25 | DE | drought | moderate | |
| PHYLLO26 | HF | watered | deep | |
| PHYLLO27 | HF | watered | deep | |
| PHYLLO28 | HF | drought | deep | |
| PHYLLO29 | HF | drought | deep | |

[1] Full field information for these two-letter abbreviations is available in the Methods section.

[2] Small indicated that the sample contained less than 233k reads, moderate indicates 233-500k reads, large indicates 500k-1.6m reads, deep indicates greater than 1.7m reads.

Functional annotations that were significant from the Dumas-Etter, TX field site (abbreviated as DE) include a range of metabolic and regulatory terms, with a large cluster of amino acid, nucleic acid, and sugar metabolic enzymes (center of Fig 2A) and a second large cluster of regulatory and response categories (top of Fig 2A), such as quorum sensing. Topics related to response to oxidative stress form a smaller cluster. Unusual categories with single small nodes include protein refolding and reactive oxygen species metabolism. The term 'transcriptional regulation' was shared with the CA term list (circled in Figs 1 and 2). The function term network (Fig 2B) also has a cluster for metal ion binding (visible at the top left of Fig 2B). After quality control, the DE site retained all six samples (Table 1) and this site had the largest number of significant annotations (Figs 2 and S1 via https://doi.org/10.5061/dryad.7m0cfxprs).

Significant annotations from the Halfway, TX field site (abbreviated as HF) include a group of biosynthetic enzymes for amino and fatty acid synthesis (Fig 3A top left), and amino acid biosynthesis enzymes (Fig 3A top right). The process annotation 'translation' was shared with the DE site (indicated by the dashed square around the node and annotation label), and amino acid transport was shared with the CA field site (indicated by a dashed diamond). In the process listing, an example unusual pathway found only in HF is self proteolysis. Functional annotations include a set of regulatory activities (e.g., kinases) and several ion binding activities. The zinc ion binding activity was shared with the DE annotation list. One unusual annotation found only in HF function was cob(*I*)yrinic acid a,c-diamide adenosyltransferase, which is part of the vitamin B12 cofactor pathway.

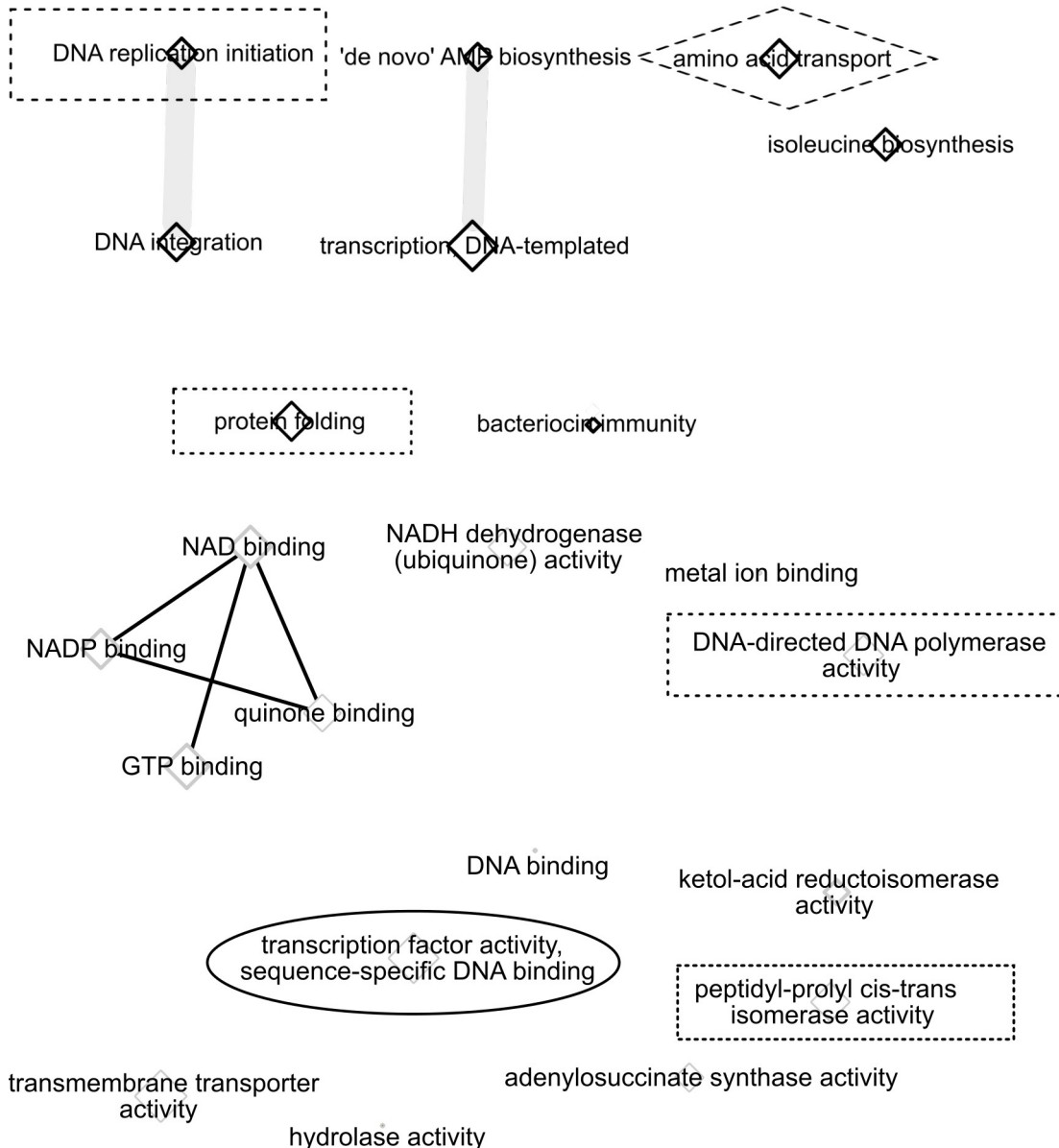

**Fig 1. Network visualization of Gene Ontology process and function annotation differences between normal water and drought treatments at the CA site.** Significant Gene Ontology (GO) annotations from ENNB analysis were grouped by semantic similarity into a network. The size of each node is proportional to the frequency of annotation relative to the GO database. Similar terms are linked with edges. Circles and boxes indicate terms shared between field sites. a) CA field site GO Process annotations that were significantly different between fully watered and drought microbial phyllosphere samples. b) CA field site GO function annotations that were significantly different between fully watered and drought microbial phyllosphere samples.

## Plant traits

To confirm that drought treatment plots were relevant for host plant performance, we analyzed plant growth measurements. All plant measurements at all sites showed significantly less growth in the treatment with less water (Fig 4). Plot effects were examined for each trait and no significant interaction between plot and replicate was found (results not shown). Mid-season plant heights were significantly less (P<0.0001) in the drought condition for the CA site. The drought-treated plants were 20% shorter, with an estimated difference between normal

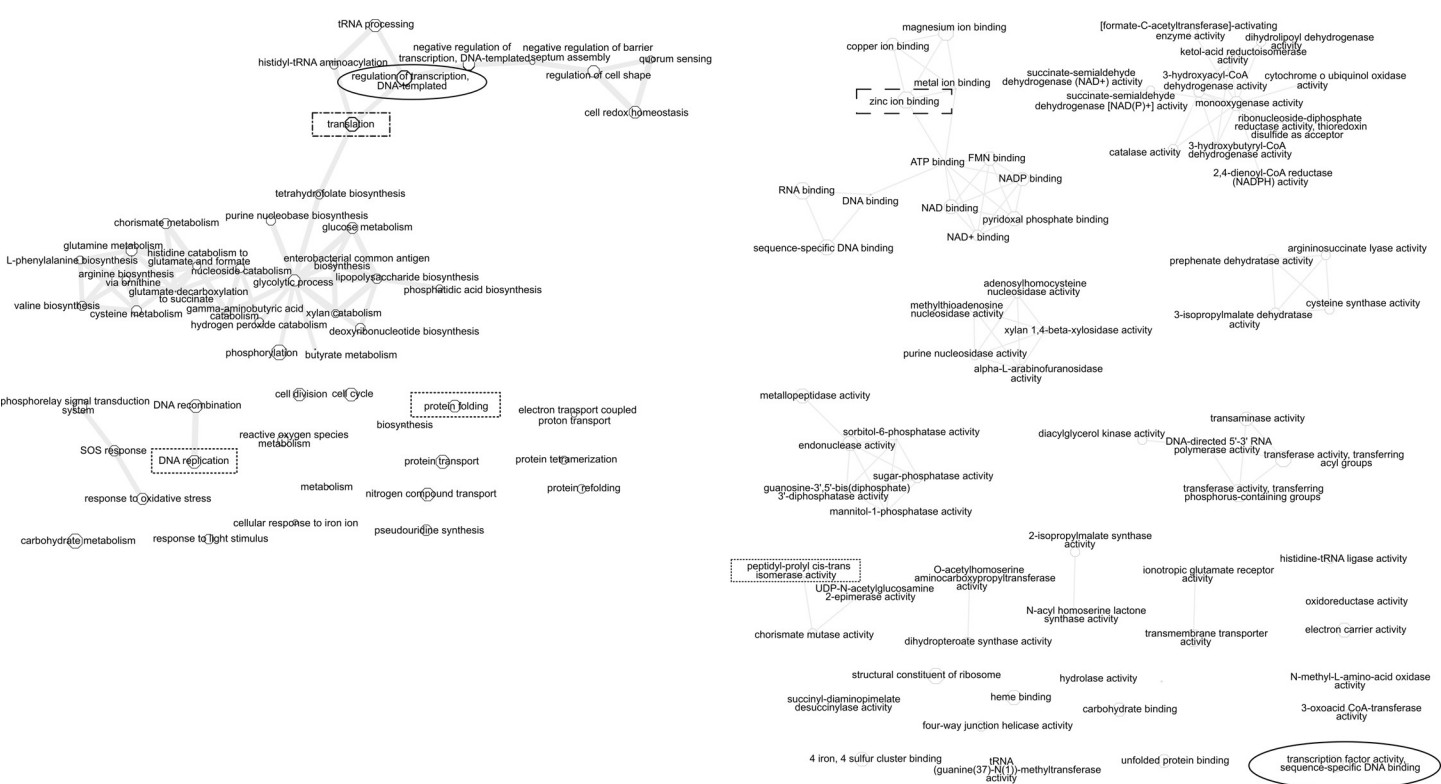

**Fig 2. Network visualization of Gene Ontology process and function annotation differences between normal water and drought treatments at the DE site.**
Significant Gene Ontology (GO) annotations from ENNB analysis were grouped by semantic similarity into a network. The size of each node is proportional to the frequency of annotation relative to the GO database. Similar terms are linked with edges. Circles and boxes indicate terms shared between field sites. a) DE field site GO Process annotations that were significantly different between fully watered and drought microbial phyllosphere samples. b) DE field site GO function annotations that were significantly different between fully watered and drought microbial phyllosphere samples.

water and drought of 0.158 meters. The DE field site with plot 101 excluded exhibited significant (P = 0.0139) effects of drought on end-of-season seed weight (Fig 4B), with the seed weights in drought reduced by about 25% (estimated difference of 0.468 grams less in drought samples). Plot 101 from the 75% site had a late-season rain event after microbiome sampling but before seed harvest that necessitated its exclusion. Drought reduced seed weight by 50% at the HF field site (Fig 4C), with P<0.0001 and an effect difference of 1.206 g less in drought seed samples. Cob diameters were also significantly smaller in the drought-treated plants (Fig 4D, 4E and 4F) with the effect size differences ranking the drought intensity of DE (1.66 mm less in drought) less than CA (2.52 mm less in drought), with the most severe cob diameter drought effects at the HF site (3.24 mm less in drought).

## Discussion

We qualitatively compared functional genes across all three sites (S1 Fig, Data Dryad repository doi: 10.5061/dryad.7m0cfxprs), though we did not fit a statistical model for comparisons of drought effects across field sites, as the field sites differed in multiple ways. There were more shared drought-treatment-relevant functional categories in comparisons of the CA and DE field sites than in comparisons with the HF site (S1 Fig). This suggests that drought severity could play a role in functional gene importance despite differences in soil and other aspects of each field environment, because the CA and DE plots did share milder drought conditions despite differences in delivery of irrigation. We would expect differences across field sites

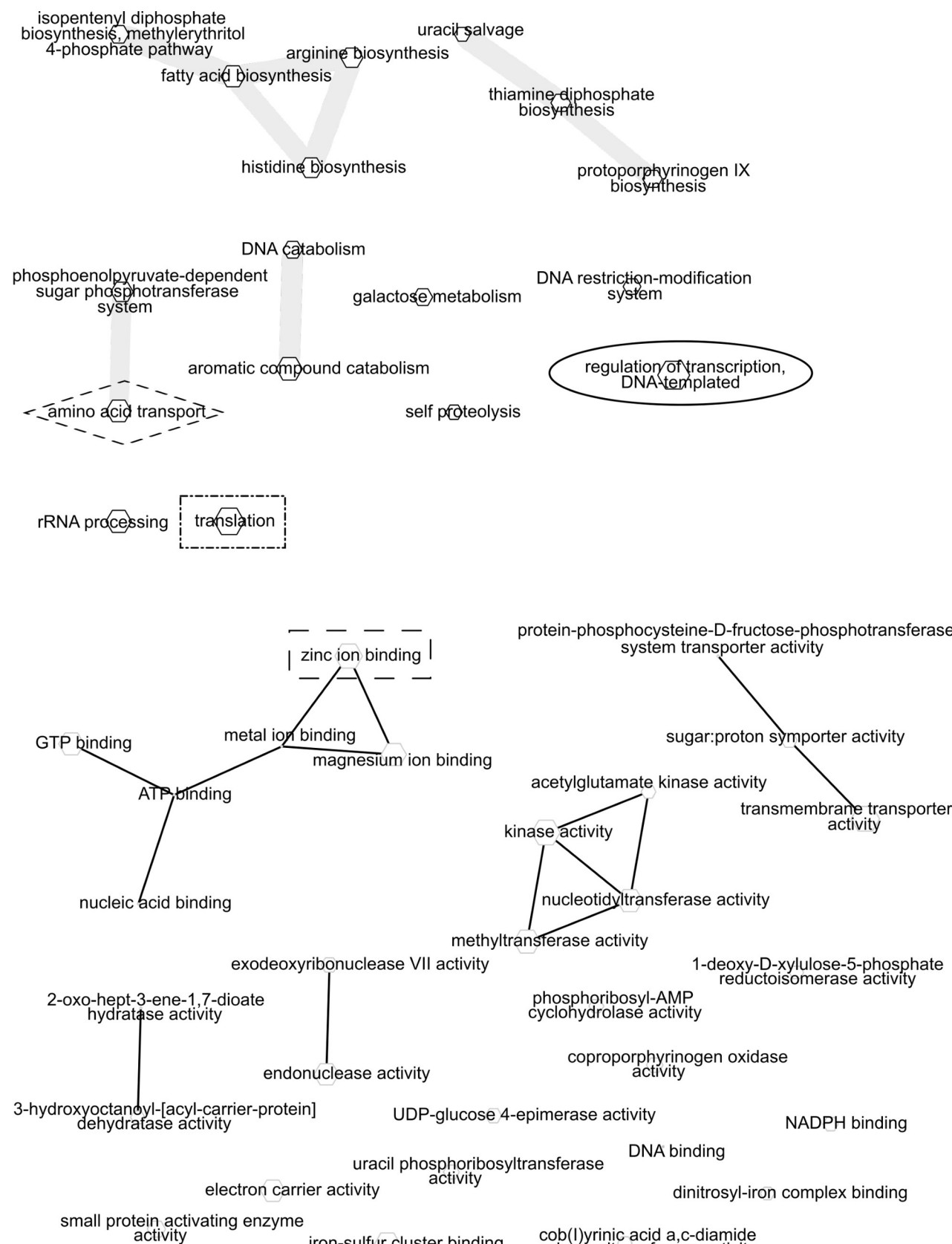

**Fig 3. Network visualization of Gene Ontology process and function annotation differences between normal water and drought treatments at the HF site.** Significant Gene Ontology (GO) annotations from ENNB analysis were grouped by semantic similarity into a network. The size of each node is proportional to the frequency of annotation relative to the GO database. Similar terms are linked with edges. Circles and boxes indicate terms shared between field sites. a) HF field site GO Process annotations that were significantly different between fully watered and drought microbial phyllosphere samples. b) HF field site GO function annotations that were significantly different between fully watered and drought microbial phyllosphere samples.

based on plant physiology and known differences in maize growth across field sites [44]. However, field site is confounded with the field-specific drought treatments in our study and we thus cannot quantitatively compare the field site annotation networks. Shared annotations across field sites often were not consistently increased or decreased in read count levels. For example, amino acid transport process read counts were higher in watered samples at the HF site and higher in drought samples at the CA site. However, the extent of drought-treatment significant annotation term sharing (without consideration of read count levels, as shown in S1 Fig) is consistent with the extent of plant growth effect, with HF sharing fewer terms and having more severe drought.

Lists of phyllosphere ribotypes from prior field studies [45–48] were used to generate a list of expected species. Expected phyllosphere species that were also present in our samples include Methylobacterium spp., Dietzia spp., and Pseudomonas spp., (Supplemental file metaphlan2.tsv in Data Dryad repository doi: 10.5061/dryad.7m0cfxprs). We carried out a detailed comparison of the annotations from the rice phyllosphere proteome [49] to our list. Six rice GO process were in the metaproteome pfam list [49], and three of the six were shared with our process lists. Recent literature on functional genes suggests that functions are more predictive than ribotype profiles [30, 31]. Therefore, in future experiments, testing of the effects of synthetic communities with similar ribosomal but different functional composition would be of broad interest. Our functional gene information is a step toward designing a future synthetic community test of functional annotation predictive ability.

In a maize leaf microbe association genetics experiment, predicted metabolic functions were more heritable than ribotypes, which also suggests that function is key [32]. Selection for specific microbial functional genes or generic markers for pathways could easily be incorporated into newer DNA-based crop genomic selection processes that are sequencing based [50–52]. The importance of incorporating microbial sequence predictors lends support to the movement toward sequencing to collect all DNA data, not just filtered SNP sets or SNPs with prior data on causality. Microbial sequences are not in linkage disequilibrium like chromosomal SNPs, so it would not be possible take advantage of tag SNPs. Because the cost of complete sequencing is decreasing, we advocate for modeling and tests of full-sequence predictors that include both host chromosomal and epiphyte functional DNA information.

We suggest that a key next step in understanding use of leaf microbial annotations for crop improvement would be to measure microbial community annotations in selected and unselected breeding program lines across multiple test sites. This would allow the estimation of the genotype and environment breeding values for functional gene annotation. That information would determine future breeding strategy and would be efficient, because collection of functional gene information could be an add-on to host breeding experiments such as g2f for maize (https://www.genomes2fields.org/) and terraRef for sorghum (http://terraref.org/). There are few publicly available field sites for drought experiments–we know of only five within the continental USA–so public-private partnerships and use of large-scale field experimental networks are logical next steps for better understanding of microbial community development for crop improvement.

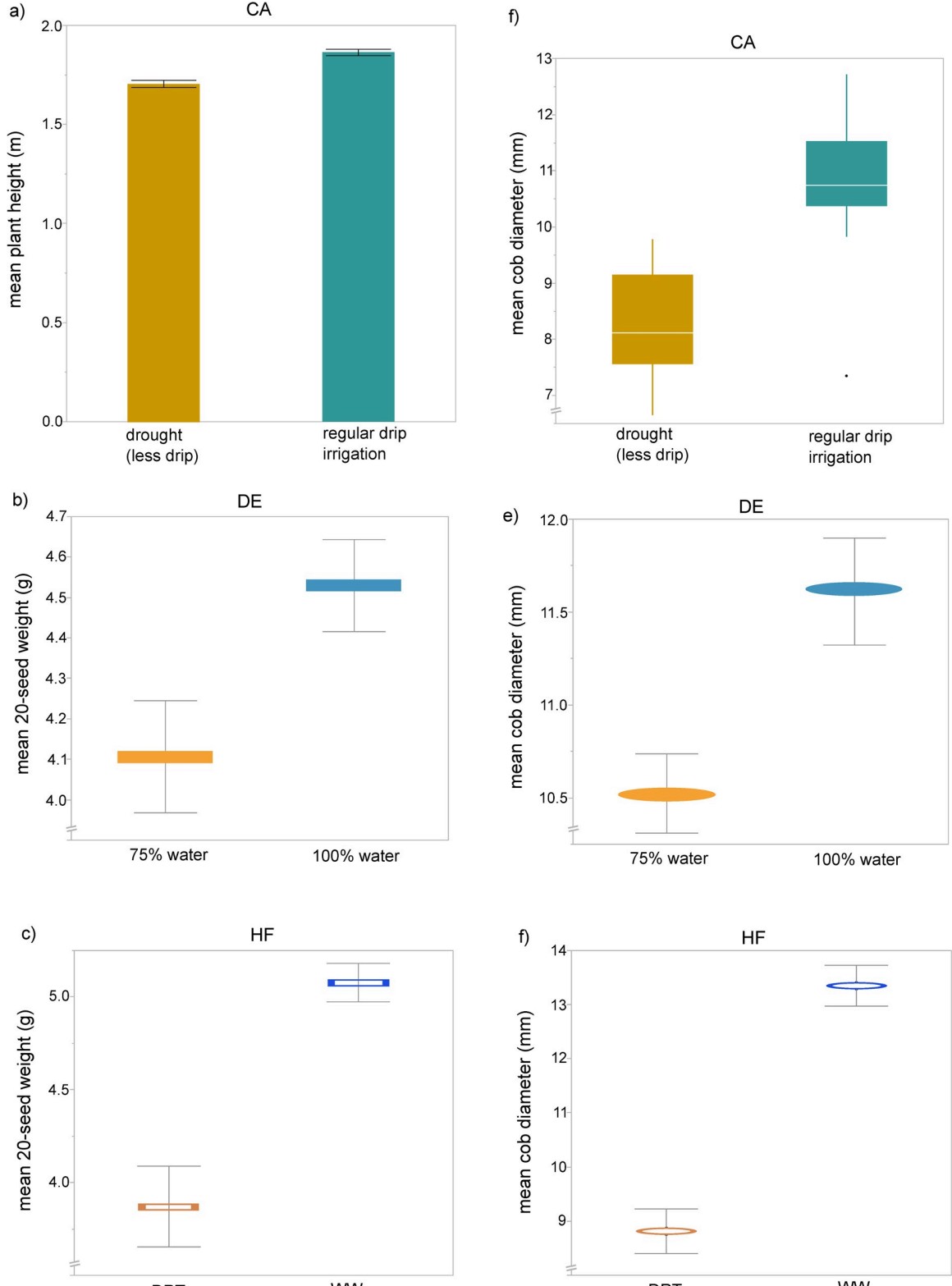

**Fig 4. Drought effects on plant growth.** Error bars are standard error and colors are grouped within a field site. a) Comparison of plant heights in drought and well-watered plots from the California-Albany (CA) field site; bar heights indicate average height in meters. Drought (less drip irrigation) n = 40, well-watered regular drip irrigation n = 40. b) Comparison of seed weights from mild drought

(75% of normal irrigation) and well-watered (100% irrigation) field blocks at the Texas-Dumas-Etter (DE) field site. Colored bar indicates mean value. Drought n = 18, well-watered n = 17. c) Comparison of seed weights from in intense drought (half of normal watering level, DRT) and well-watered (WW) field plots at the Texas-Halfway (HF) site. Colored bar indicates mean value. DRT n = 4 and WW n = 22. Zeros (cobs with no seeds from DRT) were not included in the analysis. d) Box plot of cob diameter of CA samples; white line is mean and quantiles are indicated by the box and whiskers, n = 12. e) Comparison of cob diameter by water treatment in samples from the DE site. Colored bar indicates mean value, n(75%) = 28, n(100%) = 26. f) Comparison of cob diameter by water treatment in samples from the HF site. Colored bar indicates mean value, n(DRT) = 10, n(WW) = 22.

Leaf epiphytes have short and long term intervention possibilities. Indirect selection for host effects is likely to be more cost-effective than inoculation, but that takes much longer to implement through the required multiple breeding cycles. Leaf microbes are typically not in seeds and thus not consumed. Thus, these microbes are logical targets for improved forage quality, energy extraction from biomass, or optimization of soil fertility for the next season as well as for plant host benefit.

We advocate for future experiments that build on the functional genes we identified and combining synthetic community development approaches with breeding experiments to generate knowledge that would be needed for future holobiont breeding system development. Our results allow prioritization of specific gene function pathways in choosing culturable microbe mixtures for future experiments on design of drought tolerant epiphytic communities.

## Conclusion

We identified a range of biosynthetic and regulatory microbial functional and process annotations that differed between drought and well-watered maize leaf epiphytic communities at three different field sites. These functions now provide a target for selection of beneficial microbes and for design of synthetic community casual tests of community interactions.

## Supporting information

**S1 Fig.**
(TXT)

## Acknowledgments

We thank Neha Gupta, Bryan Frank and Kelvin Li for their work on library construction and sequencing. We are obliged to Stephen P. Talley for his hard work on the ENNB analysis of an initial annotation dataset. We very much appreciate the contributions of Sarah Hake, China Lunde and the Hake lab members, who provided field and lab space for this project and carried out the field management for us. We are grateful to Robert L. Bryden, Danielle Allery Nail, and Bonnie M. Mitchell for assistance with plant trait measurements and to Monika Bihan for assistance with the sequencing and annotations. Author Roles: Ann E. Stapleton conceived the experiment, designed and carried out the field sampling, oversaw the data analysis, and wrote the manuscript. Wenwei Xu set up the Texas field sites with drought and normal irrigation treatments, planted the experimental plots, collected trait data, and edited the manuscript. Jeffrey Roach constructed the simulations. Dave Hiltbrand analyzed the annotation count data. Stuart Gordon and Brad Goodner edited the manuscript. Barbara Methe supervised the sequencing and sequence data processing and edited the manuscript.

## Author Contributions

**Conceptualization:** Brad W. Goodner, Ann E. Stapleton.

**Data curation:** Barbara A. Methe, Jeffrey Roach, Wenwei Xu, Ann E. Stapleton.

**Formal analysis:** David Hiltbrand, Jeffrey Roach.

**Funding acquisition:** Stuart G. Gordon, Brad W. Goodner, Ann E. Stapleton.

**Investigation:** Barbara A. Methe, Wenwei Xu, Ann E. Stapleton.

**Methodology:** Barbara A. Methe, David Hiltbrand, Jeffrey Roach, Ann E. Stapleton.

**Project administration:** Barbara A. Methe, Stuart G. Gordon, Ann E. Stapleton.

**Resources:** Wenwei Xu, Ann E. Stapleton.

**Software:** David Hiltbrand, Jeffrey Roach.

**Supervision:** Barbara A. Methe, Brad W. Goodner, Ann E. Stapleton.

**Validation:** Jeffrey Roach, Brad W. Goodner.

**Visualization:** Ann E. Stapleton.

**Writing – original draft:** Stuart G. Gordon, Brad W. Goodner, Ann E. Stapleton.

**Writing – review & editing:** Barbara A. Methe, David Hiltbrand, Jeffrey Roach, Wenwei Xu, Stuart G. Gordon, Brad W. Goodner, Ann E. Stapleton.

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
