## [Decision Letter · Decision Letter 0]

24 Dec 2019

PONE-D-19-16307

Functional gene categories differentiate maize leaf drought-related microbial epiphytic communities

PLOS ONE

Dear Dr Stapleton,

Thank you for submitting your manuscript to PLOS ONE. After careful consideration, we feel that it has merit but does not fully meet PLOS ONE’s publication criteria as it currently stands. Therefore, we invite you to submit a revised version of the manuscript that addresses the points raised during the review process.

The ms “Functional Gene Categories Differentiate Maize Leaf Drought-Relate Microbial Epiphytic Communities” PONE-D-19-16307 is available at the preprint server bioRxiv since 2017  (104331) as “Functional genes that distinguish maize phyllosphere metagenomes in drought and well-watered conditions”.

The present version submitted in June 2019 was reviewed by two specialists who made a number of comments and suggestions.

The authors are asked to prepare a new version addressing the reviewer´s comments and/or to reply point by point to those comments.

Since the preprint version is from 2017, the available information has expanded. As so,  references must be updated. (Cf, for example, reviewer 1 for “culturable microorganisms”).and the text adjusted accordingly.

Also in what concerns the reference microbiome: if not possible to use a plant-associated bacteria database, give a substantiate justification.

Address (or explain to  reviewer 1) the questions raised on Bioinformatics approaches.

Validation of key genes expression by qPCR would address the suggestion by reviewer 2.

We would appreciate receiving your revised manuscript by 2020 February 10th. To enhance the reproducibility of your results, we recommend that if applicable you deposit your laboratory protocols in protocols.io, where a protocol can be assigned its own identifier (DOI) such that it can be cited independently in the future. For instructions see: http://journals.plos.org/plosone/s/submission-guidelines#loc-laboratory-protocols

We look forward to receiving your revised manuscript.

Kind regards,

Sara Amancio

Academic Editor

PLOS ONE

Additional Editor Comments:

The ms “Functional Gene Categories Differentiate Maize Leaf Drought-Relate Microbial Epiphytic Communities” PONE-D-19-16307 is available at the preprint server bioRxiv since 2017 (104331) as “Functional genes that distinguish maize phyllosphere metagenomes in drought and well-watered conditions”.

The present version submitted in June 2019 was reviewed by two specialists who made a number of comments and suggestions.

The authors are asked to prepare a new version addressing the reviewer´s comments and/or to reply point by point to those comments.

Since the preprint version is from 2017, the available information has increased. As so, a references must be updated. (Cf, for example, reviewer 1 for “culturable microorganisms”).

Also in what concerns reference microbiome: if not possible to use a plant-associated bacteria database, give a substantiate justification.

Address (or explain to the reviewer) the questions on Bioinformatics approaches.

Validation of key genes expression by qPCR would address the suggestion by reviewer 2.

Reviewers' comments:

Reviewer's Responses to Questions

**Comments to the Author**

1. Is the manuscript technically sound, and do the data support the conclusions?

Reviewer #1: No

Reviewer #2: Partly

2. Has the statistical analysis been performed appropriately and rigorously? 

Reviewer #1: No

Reviewer #2: Yes

3. Have the authors made all data underlying the findings in their manuscript fully available?

Reviewer #1: Yes

Reviewer #2: Yes

4. Is the manuscript presented in an intelligible fashion and written in standard English?

Reviewer #1: No

Reviewer #2: Yes

5. Review Comments to the Author

Reviewer #1: In the manuscript by Methe et al. the authors perform a metagenomic characterization of the maize leaf epiphytic microbial community in normally irrigated plants and plants under drought stress. The authors use a replicated design across three sites USA and find that some functions recurrently have differential abundance between watered and drought stressed plants. The authors further show plant phenotypic changes in response to drought stress.

At this point, I have some major concerns regarding the methods, and clarity of presentation, as well as some minor issues that I point out below.

Major comments

1. I commend the authors for making publicly available all the raw sequencing data and for creating synapse repository with all the code used for analysis. I was able to download all of it. However, I found that I could only see the contents of the files in synapse.org repositories after registering for an account. The account is free, but I do feel this requirement places an unnecessary barrier for potential readers. It seems like synapse.org doesn’t allow for download without an account so I would encourage the authors to consider alternatives. Finally, I couldn’t find clear documentation on synapse.org regarding the type of stats that are collected regarding data usage, and which of those stats are available to the repository owners; since I was forced to provide and email to create my account and download the repositories, I opted for creating a new email account (unrelated to my affiliation and name) in order to ensure that the confidentiality of the peer review is maintained.

2. The introduction and discussion seem a bit off-topic from the core of the paper. For example the authors spend a lot of sentences talking about the potential for synthetic communities, but all the work described is on field environments with their own natural microbial communities. No strains are isolated either so it is not directly clear how this work translate synthetic communities. On the other hand, there is no description of the phyllosphere microenvironemt; what are the defining characteristics of this microbial habitat? Are there any maize specific features that might be relevant? I think presenting more about the phyllosphere itself would help interpret the functional results much betters than it is possible now.

3. In general, I found the manuscript hard to read because of excessive jargon and complex sentences. I give one example of a very hard to parse sentence “a per-field-site analysis using factorial multivariate approaches suitable for zero-inflated annotation read count data.”. This is the only mention of Zero-Inflation in the manuscript so I actually don’t know what the authors are trying to point out. It could be greatly simplified by saying “We analyzed each site independently with method X”. I think a good deal of effort needs to go into streamlining the manuscript and ensuring that terms are defined and well utilized.

4. Line 46. The number of culturable microorganisms has greatly increased in recent years. The authors provide dated references. For example, in the plant field check doi:10.1038/s41588-017-0012-9 to see that the number is closer to 50%, and this has become apparent in many other environments as well.

5. The authors use the HUMAnN2 pipeline to map their metagenomic sequences to UniProt and their associated GO annotations. This pipeline was defined with the human microbiome in mind and so I wonder how representative the reference is for plant-associated bacteria. What proportion of the reads can be mapped with this pipeline? Did the authors tried to map to a plant-associated bacteria database instead? Also, I couldn’t really figure out which file in the 10.7303/syn12933189 repository contains the details for running this pipeline.

6. I didn’t understand the rationale for sequencing two samples with HiSeq and the rest with MiSeq. What proportion of the phyllosphere are the authors capturing? Do saturation plots show that the number of new features reaches a plateau with the amount of sequencing they do?

7. In my experience, when dealing with bacterial sequences of non-model organisms, one can typically annotate less than 50% of bacterial genes with a GO tag, and many of those annotations are very general. I think this is reflected in the overlapping features highlighted in supplementary figure 1b. I would think that using a different annotation pipeline would lead to more relevant and interpretable results. Also, what proportion of the UniProt hits had a GO term?

8. I was confused by the imputation of missing HF and CA samples (lines 166-167). What is a missing sample? Why do you need to impute? What is being imputed? I also was confused by the code at “ENNB_GeneFamilies.R” and “genefamiles_real_data.nb.html”. What I see from the code `pmap_dbl(list(.$PHYLLO28, .$PHYLLO29, .$PHYLLO30), ~ (..1 + ..2 + ..3) / 3)` is that the authors are taking the average of three samples to create a fourth sample. In the next line of code the authors use the average of four samples (including the recently imputed one) to create a new average and a fifth imputed sample. This type of average-based imputation is known to be biased and to overestimate confidence; it also does not match the main text which mentions package MI. Finally, it seems like 5 samples are being created for each group, but the table in the ““Phyllosephere Metagenomics Experimental Design.docx” suggests that there are only 3 per group.

9. The authors describe some simulations to estimate the false positive rate of their pipeline. I think more details are needed. How many simulations? Did they match the number of samples and sequencing depth? What did they use as reference to simulate reads? They mention they selected the lowest false positive rate, but what was that rate? I think a figure (probably supplemental) summarizing the results of this simulation would be a good addition. Further, many assumptions go into simulating reads and/or count data, did the authors try permutation of the real data?

10. I found the two-step method for count comparison quite confusing. The authors use a binomial elastic net (after TMM) to select features, and then use a negative binomial model in the selected features. It seems like this will affect the FDR estimation since by pre-selecting features, one ends up with an excess of small p-values from the negative binomial. Further, I am not sure how well the binomial model captures the data, why not use a poisson elastic net which is closer to what the data behaves. I also don’t understand why only a two group comparison can be performed. Any model can be defined in the design matrix of the elastic net, and so more complex designs can be used. Moreover, there are tools specifically designed to test for differences between groups that are nested by some other variable (e.g. mixed models), therefore I think it should be possible and more correct to use one of these approaches to model all the samples together.

11. I was surprised by no signal in the soil samples. Drought stress should influence soil organisms as well, do the authors have any explanation for it?

12. The authors show a handful of GO terms that are differential between watered and drought-stressed plants. Some of those terms are differential in multiple sites. However, the way the data is analyzed it is hard to determine if the differential abundance is in the same direction and if the number of overlapping terms is significant or can be explained by chance alone. Please differentiate between drought enriched and drought depleted, and provide statistical quantification of the amount of overlap. Also, are the GO terms that are differential in multiple sites driven by the same uniprot genes?

13. The authors don’t show any relationship between the plant phenotypes and the microbial composition. It is well known that drought stress causes a number of changes in the plant. And it is reasonable to hypothesize that bacteria might mediate or help cope with some of those changes. But the fact that both the microbial functional composition and some plant phenotypes change in drought stress does not really tell us if that hypothesis is true. The authors could directly test for associations between drought stress phenotypes and functional content, but in the current form one cannot draw any conclusions.

14. The authors list a number of references that suggest that functional content might be more informative than taxonomic composition. This is an interesting area for discussion but the authors present no taxonomic comparisons from their data. Therefore, we don’t know if the taxonomic differences in drought vs watered plants are weaker than the functional differences they present. I think the authors should include a taxonomic comparison (which they have readily available from their metagenome sequencing) if they want to claim that functional characterization is more important than taxonomic comparisons.

Minor comments:

1. I was able to figure out from the table in the “Phyllosephere Metagenomics Experimental Design.docx” file that there are 2-3 samples per site x treatment, but I think the precise number of samples should be in the main text as a small table of figure, maybe as part of a diagram of the experimental design.

2. The sample processing methods suggest that only epiphytes are being considered. There is nothing wrong with that, but the authors should be explicit about it in the different sections of the main text (ie. abstract, introduction, results, discussion).

3. Line 230. Threshold on what parameter?

4. For figs 1-3, I found the circles and boxes hard to follow. I suggest the authors try colors since those are more visually obvious. In any case, I also think the authors should include a graphical legend indicating what the different symbols/sizes/colors mean.

Reviewer #2: The manuscript “Functional gene categories differentiate maize leaf drought-related microbial epiphytic communities” is well written, and analyze the functional gene of phyllosphere microbiome associated to maiz leaf growing in two conditions, drought and well-watered. The analysis of genes in both growth conditions shows the importance of genes related with aminoacids biosynthesis and transport, metal ion binding and regulatory functions as quorum sensing. However, additional experiments as expression differential of genes involved with aminoacids byosinthesis or drought responses in maize would support the hypothesis of bioinformatic analysis on functional gene. The agronomic data showed in the figure 4b-f, the scale of “Y” axis in the graphs need correct the scale and letter size. Additional information showing the effects on plant growth maize plants under the two conditions evaluated could complement the agronomic data evaluated.

6. PLOS authors have the option to publish the peer review history of their article (what does this mean?). If published, this will include your full peer review and any attached files.

Reviewer #1: No

Reviewer #2: No

---

## [Author Response · Author response to Decision Letter 0]

12 Apr 2020

We very much appreciate these thorough, thoughtful review comments. Our replies are in plain text below the italicized comments and we have uploaded our reply as a file in this resubmission. 

Additional Editor Comments:

The ms “Functional Gene Categories Differentiate Maize Leaf Drought-Relate Microbial Epiphytic Communities” PONE-D-19-16307 is available at the preprint server bioRxiv since 2017 (104331) as “Functional genes that distinguish maize phyllosphere metagenomes in drought and well-watered conditions”.

The present version submitted in June 2019 was reviewed by two specialists who made a number of comments and suggestions.

The authors are asked to prepare a new version addressing the reviewer´s comments and/or to reply point by point to those comments.

Since the preprint version is from 2017, the available information has increased. As so, a references must be updated. (Cf, for example, reviewer 1 for “culturable microorganisms”).

We apologize for the confusion – the biorxiv preprint was an older version with different authors, title and text. We have now updated the biorxiv to the version we submitted to PLOS ONE. In this version the references are up to date.

Also in what concerns reference microbiome: if not possible to use a plant-associated bacteria database, give a substantiate justification.

We address this by explaining the admittedly confusing name of the microbiome resource; the developers do note on the first page of their documentation that the name is a historical artifact and is no longer an accurate descriptor of the software pipeline.

Address (or explain to the reviewer) the questions on Bioinformatics approaches.

We address each point in the reviewer comments below. 

Validation of key genes expression by qPCR would address the suggestion by reviewer 2.

We explain in the reply to reviewer 2 that this is not a good option as our experimental design is not suitable for testing for predictive gene expression change (which occurs on short time scales). Our focus is on gene family DNA content differences that reflect microbial community differences across drought factor levels. 

We have complied with these format instructions. We have a latex version of our manuscript and we would prefer to submit using that format if possible. We also have created a docx format with track changes as requested.

Reviewer #1: In the manuscript by Methe et al. the authors perform a metagenomic characterization of the maize leaf epiphytic microbial community in normally irrigated plants and plants under drought stress. The authors use a replicated design across three sites USA and find that some functions recurrently have differential abundance between watered and drought stressed plants. The authors further show plant phenotypic changes in response to drought stress.

At this point, I have some major concerns regarding the methods, and clarity of presentation, as well as some minor issues that I point out below.

Major comments

1. I commend the authors for making publicly available all the raw sequencing data and for creating synapse repository with all the code used for analysis. I was able to download all of it. However, I found that I could only see the contents of the files in synapse.org repositories after registering for an account. The account is free, but I do feel this requirement places an unnecessary barrier for potential readers. It seems like synapse.org doesn’t allow for download without an account so I would encourage the authors to consider alternatives. Finally, I couldn’t find clear documentation on synapse.org regarding the type of stats that are collected regarding data usage, and which of those stats are available to the repository owners; since I was forced to provide and email to create my account and download the repositories, I opted for creating a new email account (unrelated to my affiliation and name) in order to ensure that the confidentiality of the peer review is maintained.

We agree that the synapse repository rules, which were put into place to ensure that users agreed to the data conduct standards, do not serve anonymous reviewers well. We appreciate all the effort you put into accessing our code and documentation! We used synapse.org as it is free, version-controlled, and allowed enough space for all our files, data and documentation. Other options did not meet all of those constraints for us at that time. We will make our synapse repositories public and freely accessible upon publication of our manuscript.

2. The introduction and discussion seem a bit off-topic from the core of the paper. For example the authors spend a lot of sentences talking about the potential for synthetic communities, but all the work described is on field environments with their own natural microbial communities. No strains are isolated either so it is not directly clear how this work translate synthetic communities. On the other hand, there is no description of the phyllosphere microenvironemt; what are the defining characteristics of this microbial habitat? Are there any maize specific features that might be relevant? I think presenting more about the phyllosphere itself would help interpret the functional results much betters than it is possible now.

There may have been some confusion between the old preprint and the new PLOS submission. We agree that it is confusing to have the old preprint visible (with the prior title and different author list), so we have updated biorxiv with the version we submitted to PLOS. That version is now available from biorxiv. We only discuss synthetic communities as a future research area in the discussion section of the updated current PLOS manuscript. 

We did not measure any specific leaf habitat features in this work, so we did not discuss that literature. We have examined leaf surfaces via scanning electron microscopy in some previous work (https://f1000research.com/articles/6-1698), but we did not feel it was necessary to cite this. We would be happy to add this self-citation or other cites to habitat features, but it is not a measured part of the experimental results that we present. 

3. In general, I found the manuscript hard to read because of excessive jargon and complex sentences. I give one example of a very hard to parse sentence “a per-field-site analysis using factorial multivariate approaches suitable for zero-inflated annotation read count data.”. This is the only mention of Zero-Inflation in the manuscript so I actually don’t know what the authors are trying to point out. It could be greatly simplified by saying “We analyzed each site independently with method X”. I think a good deal of effort needs to go into streamlining the manuscript and ensuring that terms are defined and well utilized.

We have added additional explanation to the methods section.

4. Line 46. The number of culturable microorganisms has greatly increased in recent years. The authors provide dated references. For example, in the plant field check doi:10.1038/s41588-017-0012-9 to see that the number is closer to 50%, and this has become apparent in many other environments as well.

We reworded that sentence (line 45) to be up-to date and to be less specific about culturability numbers, as they certainly can change over time. The doi (doi:10.1038/s41588-017-0012-9) refers to an analysis of plant roots and does not specifically describe culturability experiments as far as we can tell from the abstract; we do not have access to the full text. 

5. The authors use the HUMAnN2 pipeline to map their metagenomic sequences to UniProt and their associated GO annotations. This pipeline was defined with the human microbiome in mind and so I wonder how representative the reference is for plant-associated bacteria. What proportion of the reads can be mapped with this pipeline? Did the authors tried to map to a plant-associated bacteria database instead? Also, I couldn’t really figure out which file in the 10.7303/syn12933189 repository contains the details for running this pipeline.

This pipeline “is appropriate for any type of microbial community shotgun sequence profiling (i.e. not just the human microbiome; the name is a historical product of its origin” as stated by the pipeline developer at https://bitbucket.org/biobakery/biobakery/wiki/humann2. 

6. I didn’t understand the rationale for sequencing two samples with HiSeq and the rest with MiSeq. What proportion of the phyllosphere are the authors capturing? Do saturation plots show that the number of new features reaches a plateau with the amount of sequencing they do?

Sequence data from a sample are inherently a population sample, as we cannot plan to sequence every DNA molecule in a sample using this technology. This ‘sampling of molecules’ aspect of sequencing was an important criteria in our data analysis design; we use zero-inflated distributions to appropriately model these data. Rather than rely on saturation plots and ‘equal samples’ we modeled the output reads using a more appropriate distribution and an analysis method that allows for any differences in feature detection. We did use two different sequencing technologies to check for any biases in our quality control processing from the raw reads; we did not observe any consistent differences in QC parameters between the MiSeq and HiSeq outputs.

7. In my experience, when dealing with bacterial sequences of non-model organisms, one can typically annotate less than 50% of bacterial genes with a GO tag, and many of those annotations are very general. I think this is reflected in the overlapping features highlighted in supplementary figure 1b. I would think that using a different annotation pipeline would lead to more relevant and interpretable results. Also, what proportion of the UniProt hits had a GO term?

We chose GO as this annotation type is widely used and interpretable. One reason we made all our raw data and intermediate files available is to allow others to apply updated functional annotations if desired. It would be useful in the future to do a simulation study of the inherent error and distributions in functional annotations, though this kind of simulation would be a more theoretical and statistical aspect of data analysis and thus is not part of our experimental study. 

8. I was confused by the imputation of missing HF and CA samples (lines 166-167). What is a missing sample? Why do you need to impute? What is being imputed? I also was confused by the code at “ENNB_GeneFamilies.R” and “genefamiles_real_data.nb.html”. What I see from the code `pmap_dbl(list(.$PHYLLO28, .$PHYLLO29, .$PHYLLO30), ~ (..1 + ..2 + ..3) / 3)` is that the authors are taking the average of three samples to create a fourth sample. In the next line of code the authors use the average of four samples (including the recently imputed one) to create a new average and a fifth imputed sample. This type of average-based imputation is known to be biased and to overestimate confidence; it also does not match the main text which mentions package MI. Finally, it seems like 5 samples are being created for each group, but the table in the ““Phyllosephere Metagenomics Experimental Design.docx” suggests that there are only 3 per group.

We clarified our methods data analysis section and our code repository to more clearly explain how we used the imputations to get the lambda value that was used in the cross validation for the elastic net portion of the ENNB analysis of the sample data. The simulations were used to select appropriate parameters and P-value thresholds for ENNB to ensure optimal error control in our analysis of the sample data. We added more explanation of these steps to our code repository, and we re-confirmed that the R code runs and produces the expected outputs. We agree that this was poorly explained in the original text, and we appreciate your careful review of our code and documentation.

9. The authors describe some simulations to estimate the false positive rate of their pipeline. I think more details are needed. How many simulations? Did they match the number of samples and sequencing depth? What did they use as reference to simulate reads? They mention they selected the lowest false positive rate, but what was that rate? I think a figure (probably supplemental) summarizing the results of this simulation would be a good addition. Further, many assumptions go into simulating reads and/or count data, did the authors try permutation of the real data?

Permutation of real data is not appropriate for data analysis from our small-n experimental design. We need to choose analysis parameters that have an optimal true and false positive level, and thus we must create known-truth simulations. Permutations would address a different question and are quite problematic for small numbers of highly multivariate data with zero-inflated distributions. The R notebook in the supplemental data has a succinct summary of the simulation analyses that were used, with a discussion of how method performance degrades when genes are not assigned to the correct gene family by sequence comparison. 

10. I found the two-step method for count comparison quite confusing. The authors use a binomial elastic net (after TMM) to select features, and then use a negative binomial model in the selected features. It seems like this will affect the FDR estimation since by pre-selecting features, one ends up with an excess of small p-values from the negative binomial. Further, I am not sure how well the binomial model captures the data, why not use a poisson elastic net which is closer to what the data behaves. I also don’t understand why only a two group comparison can be performed. Any model can be defined in the design matrix of the elastic net, and so more complex designs can be used. Moreover, there are tools specifically designed to test for differences between groups that are nested by some other variable (e.g. mixed models), therefore I think it should be possible and more correct to use one of these approaches to model all the samples together.

The true and false positive rates in the ENNB two-stage approach are described at length in their original paper (Pookhao et al, Bioinformatics 2015); the method parameters were carefully calibrated to avoid excess false positives. We chose this ENNB method specifically because it models the count data appropriately, with a well-accepted distribution for counts, the zero-inflated negative binomial. We know that our data type is zero-inflated count data, and thus it is the appropriate distribution to use. The ENNB package only supports two-group (single factor) comparison. We do not know of any packages that allow hierarchical factor analysis (as with our experimental design that has the drought factor nested in location) with zero-inflated count data. Additional details on the complexity of this modeling problem are described in papers such as those from D. Witten and colleagues (https://amstat.tandfonline.com/doi/abs/10.1080/10618600.2015.1067217#.XijgYhf_qkY). We also explain the experimental design limits across geographies in our reply to comment 12 below.

11. I was surprised by no signal in the soil samples. Drought stress should influence soil organisms as well, do the authors have any explanation for it?

The extraction method was designed to be specific for leaf epiphytes, not for soils. Thus, we expected little DNA from these samples -- and that is what we observed. The soil samples control for the possible confounding effect of having soil particles present on leaf surfaces and inadvertently extracting soil-associated microbial DNA. 

12. The authors show a handful of GO terms that are differential between watered and drought-stressed plants. Some of those terms are differential in multiple sites. However, the way the data is analyzed it is hard to determine if the differential abundance is in the same direction and if the number of overlapping terms is significant or can be explained by chance alone. Please differentiate between drought enriched and drought depleted, and provide statistical quantification of the amount of overlap. Also, are the GO terms that are differential in multiple sites driven by the same uniprot genes?

We purposefully did not do a statistical analysis across the geographic locations, as there are many factors that differ across geographies and three sites are thus a priori confounded across these multiple differences. The goal of using multiple sites is to illustrate that functional gene changes are likely to be site-specific, and to encourage follow-up large-scale experiments or future experimental designs such as transplantation of soils and plants across field sites. 

13. The authors don’t show any relationship between the plant phenotypes and the microbial composition. It is well known that drought stress causes a number of changes in the plant. And it is reasonable to hypothesize that bacteria might mediate or help cope with some of those changes. But the fact that both the microbial functional composition and some plant phenotypes change in drought stress does not really tell us if that hypothesis is true. The authors could directly test for associations between drought stress phenotypes and functional content, but in the current form one cannot draw any conclusions.

This would require a ‘multi-view’ analysis; this type of analysis is of strong recent research interest (for example, https://academic.oup.com/biostatistics/advance-article-abstract/doi/10.1093/biostatistics/kxz001/5310125). However, there is little clarity on optimal methods for such an analysis at this time. We prefer the normal statistical approach of considering all measurements on a sample to be multivariate. Multivariate analysis of a factorial experimental design does not allow inference on the order of events; though the plant measurements are taken later in the season the analysis is more appropriately considered as one set of multiple measurements of the same sampling unit. 

14. The authors list a number of references that suggest that functional content might be more informative than taxonomic composition. This is an interesting area for discussion but the authors present no taxonomic comparisons from their data. Therefore, we don’t know if the taxonomic differences in drought vs watered plants are weaker than the functional differences they present. I think the authors should include a taxonomic comparison (which they have readily available from their metagenome sequencing) if they want to claim that functional characterization is more important than taxonomic comparisons.

We provide our experimental data (including the taxonomic assignments) for anyone interested in using real data to examine this question. In our view, the relative value of taxonomic information for a range of different microbial community questions and for different types of experimental designs is best studied using simulations and large data sets, rather than using the focused experimental designs that we used to address functional differences with and without drought. As you note, we cite these references to support our focus on functional analysis, not as part of a different kind of study on the relative usefulness of taxonomic versus functional annotations.

Minor comments:

1. I was able to figure out from the table in the “Phyllosephere Metagenomics Experimental Design.docx” file that there are 2-3 samples per site x treatment, but I think the precise number of samples should be in the main text as a small table of figure, maybe as part of a diagram of the experimental design.

We appreciate this suggestion but prefer to keep the larger table with full experimental details in the supplemental files to keep the number of figures and tables in the main text more compact. 

2. The sample processing methods suggest that only epiphytes are being considered. There is nothing wrong with that, but the authors should be explicit about it in the different sections of the main text (ie. abstract, introduction, results, discussion).

This is described in the title of the article as well as in these sections. Perhaps there was confusion between the submitted version of the article and the old bioarxiv preprint?

3. Line 230. Threshold on what parameter?

Corrected. 

4. For figs 1-3, I found the circles and boxes hard to follow. I suggest the authors try colors since those are more visually obvious. In any case, I also think the authors should include a graphical legend indicating what the different symbols/sizes/colors mean.

We chose the gray-scales, shapes and patterns to be color-blind friendly. We agree that color-coding would be useful for many readers, but we also wish to ensure access for all. If PLOS ONE allows a graphical legend in our case we would be happy to include that within the figures. 

Reviewer #2: The manuscript “Functional gene categories differentiate maize leaf drought-related microbial epiphytic communities” is well written, and analyze the functional gene of phyllosphere microbiome associated to maiz leaf growing in two conditions, drought and well-watered. The analysis of genes in both growth conditions shows the importance of genes related with aminoacids biosynthesis and transport, metal ion binding and regulatory functions as quorum sensing. However, additional experiments as expression differential of genes involved with aminoacids byosinthesis or drought responses in maize would support the hypothesis of bioinformatic analysis on functional gene. 

Our experiment was not designed to detect RNA expression differences, which occur on a short time scale and would require much more intensive sampling and a different experimental design. We agree that follow-up experiments should consider appropriate experimental designs for measurement of gene expression if there is future interest in short-term adaptive processes in epiphytic microbiomes.

The agronomic data showed in the figure 4b-f, the scale of “Y” axis in the graphs need correct the scale and letter size. 

We have a high-resolution figure with appropriate text size available; unfortunately it could not be submitted in the journal’s MSS submission system. We will work with the editors to ensure that the final figure is of appropriate quality. 

Additional information showing the effects on plant growth maize plants under the two conditions evaluated could complement the agronomic data evaluated.

To confirm that the drought conditions we applied in our experiments were relevant, we measured plant traits normally used in agronomic experiments, such as plant height and seed weight. This measurement met our goal of checking that the drought conditions were agronomically relevant.

---

## [Decision Letter · Decision Letter 1]

6 May 2020

PONE-D-19-16307R1

Functional gene categories differentiate maize leaf drought-related microbial epiphytic communities

PLOS ONE

Dear Dr. Stapleton,

Thank you for submitting your manuscript to PLOS ONE. After careful consideration, we feel that it has merit but does not fully meet PLOS ONE’s publication criteria as it currently stands. Therefore, we invite you to submit a revised version of the manuscript that addresses the points raised during the review process.

The ms “Functional Gene Categories Differentiate Maize Leaf Drought-Relate Microbial Epiphytic Communities” PONE-D-19-16307R1 was resubmitted in april 2020, almost four months after the decision of the reviewing of the first version was sent to authors.

R1 version was reviewed by the same experts as the first version. Both asked for further revision and addressed comments and questions to the present version.

Besides other points raised by the reviewers, soil sampling and statistical analysis must be thoroughly addressed.

The authors are asked to prepare a new version addressing the reviewer´s comments and/or to reply point by point to those comments.

We would appreciate receiving your revised manuscript by 15th June. To enhance the reproducibility of your results, we recommend that if applicable you deposit your laboratory protocols in protocols.io, where a protocol can be assigned its own identifier (DOI) such that it can be cited independently in the future. For instructions see: http://journals.plos.org/plosone/s/submission-guidelines#loc-laboratory-protocols

We look forward to receiving your revised manuscript.

Kind regards,

Sara Amancio

Academic Editor

PLOS ONE

Reviewers' comments:

Reviewer's Responses to Questions

**Comments to the Author**

1. If the authors have adequately addressed your comments raised in a previous round of review and you feel that this manuscript is now acceptable for publication, you may indicate that here to bypass the “Comments to the Author” section, enter your conflict of interest statement in the “Confidential to Editor” section, and submit your "Accept" recommendation.

Reviewer #1: (No Response)

Reviewer #2: All comments have been addressed

2. Is the manuscript technically sound, and do the data support the conclusions?

Reviewer #1: No

Reviewer #2: Partly

3. Has the statistical analysis been performed appropriately and rigorously? 

Reviewer #1: No

Reviewer #2: Yes

4. Have the authors made all data underlying the findings in their manuscript fully available?

Reviewer #1: Yes

Reviewer #2: Yes

5. Is the manuscript presented in an intelligible fashion and written in standard English?

Reviewer #1: Yes

Reviewer #2: Yes

6. Review Comments to the Author

Reviewer #1: The revised manuscript by Methe et al. makes some clarifications regarding various points, but I think it still lacks enough detail in a number of aspects. I also have some methodological comments.

1. I am still surprised by the lack of soil sequences. The authors write in their response that methods were specifically designed for leaf epiphytes but there is no clear methodological description of how soil samples were generated. DNA extraction with PowerSoil would certainly produce a high number of microbial DNA from soil samples if standard soil sampling methods were used. Please include a description of the precise soil sampling method.

2. The authors of the HUMaN2 pipeline do state in their website that their method can be used for any community. However, the HUMaN2 publication (doi: 10.1038/s41592-018-0176-y) makes no such claim and only provides analysis for human and marine communities. The methods may be general but since it is a reference based approach, it will always be sensitive to the reference database. Therefore, I urge great caution when interpreting results from this approach on plant-associated communities.

3. Line 190. I had asked about the false discovery rate (FDR) and the author state that they chose the lowest possible false positive rate (FPR) but they never state what was that rate. I think the final FDR/FPR ratio needs to be reported in the manuscript otherwise one doesn’t know how ot interpret the results. Also, FPR and FDR are not identical and the authors should control via FDR not FPR.

4. line 220 states that “little correlation between depth of microbial sequence and annotation quality (Table 1).”, but the table doesn’t actually shows that. It just tells me that some samples were sequenced different amounts, but there is no quality metrics.

5. Lines 295-297 state that there is a correlation between drought severity and similarity in microbial functional changes, but with only three sites and without a statistical analysis to back it, I don’t see how that apparent correlation can be interpreted.

6. From the title abstract and conclusion it seems as if as if functional categories consistently differentiate between drought and well-watered conditions; however, in their response the authors state that “The goal of using multiple sites is to illustrate that functional gene changes are likely to be site-specific”. I think the authors need to be more clear about what conclusion they are trying to present. Moreover, claiming site specific differences requires comparisons between sites which the authors refrain from because of lack of appropriate statistical tools. I think this is a valid argument, but then the authors need to be explicit about the limitations of their conclusions. One can make claims about specific sites, but it is impossible to claim if the differences between sites are particularly big or small because we don’t know what to expect.

Reviewer #2: Why the agronomic data found in figure 4 was drawn from different sites? For example a) plant height of the California-Albany (CA) site, b) Seed weight of the Texas-Dumas-Etter site, c) Seed weight of the Texas-Halfway site, d) stem diameter of the California site -Albany ... The same agronomic data record was made for the different sites, because the comparison between the sites in the different agronomic variables analyzed is not illustrated.

Ideally, the effects of affected plants on the effects of plants against drought can be shown, images of plants grown under both conditions, perhaps modify Figure 4 or attach it as supplementary material.

Why use different degrees of drought between different sites and they were not the same at all sites?

How to consider the height of the plants, at what height of the plant to take the diameter of the plant, the weight of the seeds of how many seeds and in the condition raised? It is not described in materials and methods or if the strategy was taken from previous work.

The sizes of the graphics and letters of the axes are small, they need to be edited and the figures must be loaded according to the PLOS ONE guidelines.

7. PLOS authors have the option to publish the peer review history of their article (what does this mean?). If published, this will include your full peer review and any attached files.

Reviewer #1: Yes: Sur Herrera Paredes

Reviewer #2: No

---

## [Author Response · Author response to Decision Letter 1]

20 Jun 2020

May 20, 2020

Reply to reviewer 1 comments. Reviewer requests are in italics, and our replies to reviewer 1 are in plain text.

Reviewer's Responses to Questions

Comments to the Author

1. If the authors have adequately addressed your comments raised in a previous round of review and you feel that this manuscript is now acceptable for publication, you may indicate that here to bypass the “Comments to the Author” section, enter your conflict of interest statement in the “Confidential to Editor” section, and submit your "Accept" recommendation.

Reviewer #1: (No Response)

Reviewer #2: All comments have been addressed

 We provide replies to reviewer 1’s comments below.

6. Review Comments to the Author

Reviewer #1: The revised manuscript by Methe et al. makes some clarifications regarding various points, but I think it still lacks enough detail in a number of aspects. I also have some methodological comments.

1. I am still surprised by the lack of soil sequences. The authors write in their response that methods were specifically designed for leaf epiphytes but there is no clear methodological description of how soil samples were generated. DNA extraction with PowerSoil would certainly produce a high number of microbial DNA from soil samples if standard soil sampling methods were used. Please include a description of the precise soil sampling method.

Since the goal was to check the extent of soil microbial DNA on leaves due to soil particles on the leaf surface, the soil samples were extracted using the same method that was used for the leaf epiphyte samples (placing soil samples taken adjacent to plants into sterile water with Silwet, filtering, and extracting). We added an additional sentence in the methods to explain this more clearly.

2. The authors of the HUMaN2 pipeline do state in their website that their method can be used for any community. However, the HUMaN2 publication (doi: 10.1038/s41592-018-0176-y) makes no such claim and only provides analysis for human and marine communities. The methods may be general but since it is a reference based approach, it will always be sensitive to the reference database. Therefore, I urge great caution when interpreting results from this approach on plant-associated communities.

We added additional explanation of the fact that this is a reference-based approach to the methods section.

3. Line 190. I had asked about the false discovery rate (FDR) and the author state that they chose the lowest possible false positive rate (FPR) but they never state what was that rate. I think the final FDR/FPR ratio needs to be reported in the manuscript otherwise one doesn’t know how ot interpret the results. Also, FPR and FDR are not identical and the authors should control via FDR not FPR.

We added additional explanation to the main manuscript, with information about where in the supplemental methods additional information such as the confusion matrices may be found. We created groups (A, B, C) with sequences from the same gene family and different gene families; A and B were low-similarity, while A and C were typical higher similarity levels. When we compared different combinations of groups, AB, BC, AC, the FDRs after running through the ENNB pipeline with TMM normalization were 0.846, 0.195, and 0.088 respectively using the threshold of 0.001 to compare p-values. The goal for the simulations to determine if ENNB was a viable method for detecting gene counts between groups. The low FDR calculated using simulations on groups A and C allowed us to determine that ENNB would be an acceptable tool to run against the real data provided the sequence match value to declare function similarity was set to the higher level. The notebook is genefamilies_simulations.Rmd.

4. line 220 states that “little correlation between depth of microbial sequence and annotation quality (Table 1).”, but the table doesn’t actually shows that. It just tells me that some samples were sequenced different amounts, but there is no quality metrics.

We corrected this sentence, thank you for pointing this out.

5. Lines 295-297 state that there is a correlation between drought severity and similarity in microbial functional changes, but with only three sites and without a statistical analysis to back it, I don’t see how that apparent correlation can be interpreted.

We agree that comparisons across sites are qualitative, and ‘correlation’ in this sentence is a descriptive term, not a specific measurement.

6. From the title abstract and conclusion it seems as if as if functional categories consistently differentiate between drought and well-watered conditions; however, in their response the authors state that “The goal of using multiple sites is to illustrate that functional gene changes are likely to be site-specific”. I think the authors need to be more clear about what conclusion they are trying to present. Moreover, claiming site specific differences requires comparisons between sites which the authors refrain from because of lack of appropriate statistical tools. I think this is a valid argument, but then the authors need to be explicit about the limitations of their conclusions. One can make claims about specific sites, but it is impossible to claim if the differences between sites are particularly big or small because we don’t know what to expect.

We edited the conclusion to clarify this point.

Reviewer #2: Why the agronomic data found in figure 4 was drawn from different sites? For example a) plant height of the California-Albany (CA) site, b) Seed weight of the Texas-Dumas-Etter site, c) Seed weight of the Texas-Halfway site, d) stem diameter of the California site -Albany ... The same agronomic data record was made for the different sites, because the comparison between the sites in the different agronomic variables analyzed is not illustrated.

Ideally, the effects of affected plants on the effects of plants against drought can be shown, images of plants grown under both conditions, perhaps modify Figure 4 or attach it as supplementary material.

Why use different degrees of drought between different sites and they were not the same at all sites?

How to consider the height of the plants, at what height of the plant to take the diameter of the plant, the weight of the seeds of how many seeds and in the condition raised? It is not described in materials and methods or if the strategy was taken from previous work.

The sizes of the graphics and letters of the axes are small, they need to be edited and the figures must be loaded according to the PLOS ONE guidelines.

We have uploaded our correctly formatted figure as an eps file.

---

## [Decision Letter · Decision Letter 2]

29 Jul 2020

Functional gene categories differentiate maize leaf drought-related microbial epiphytic communities

PONE-D-19-16307R2

Dear Dr. Stapleton

We’re pleased to inform you that your manuscript has been judged scientifically suitable for publication and will be formally accepted for publication once it meets all outstanding technical requirements.

Kind regards,

Sara Amancio

Academic Editor

PLOS ONE

Additional Editor Comments (optional):

Reviewers' comments:

Reviewer's Responses to Questions

**Comments to the Author**

1. If the authors have adequately addressed your comments raised in a previous round of review and you feel that this manuscript is now acceptable for publication, you may indicate that here to bypass the “Comments to the Author” section, enter your conflict of interest statement in the “Confidential to Editor” section, and submit your "Accept" recommendation.

Reviewer #1: All comments have been addressed

Reviewer #2: All comments have been addressed

2. Is the manuscript technically sound, and do the data support the conclusions?

Reviewer #1: Yes

Reviewer #2: Partly

3. Has the statistical analysis been performed appropriately and rigorously? 

Reviewer #1: Yes

Reviewer #2: Yes

4. Have the authors made all data underlying the findings in their manuscript fully available?

Reviewer #1: Yes

Reviewer #2: Yes

5. Is the manuscript presented in an intelligible fashion and written in standard English?

Reviewer #1: Yes

Reviewer #2: Yes

6. Review Comments to the Author

Reviewer #1: (No Response)

Reviewer #2: Is not clear why used agronomic data from different sampling site for discuss your results in the figure 4, like height, cob diameter, seed biomass. Do you have some pictures that ilustres the effects on plant phenotype?

7. PLOS authors have the option to publish the peer review history of their article (what does this mean?). If published, this will include your full peer review and any attached files.

Reviewer #1: **Yes: **Sur Herrera Paredes

Reviewer #2: No

---

## [Editor Report · Acceptance letter]

17 Aug 2020

PONE-D-19-16307R2 

Functional gene categories differentiate maize leaf drought-related microbial epiphytic communities 

Dear Dr. Stapleton:

I'm pleased to inform you that your manuscript has been deemed suitable for publication in PLOS ONE. Congratulations! Your manuscript is now with our production department. 

Kind regards, 

on behalf of

Prof Sara Amancio 

Academic Editor

PLOS ONE